# Flaviviruses in AntiTumor Therapy

**DOI:** 10.3390/v15101973

**Published:** 2023-09-22

**Authors:** Alina S. Nazarenko, Mikhail F. Vorovitch, Yulia K. Biryukova, Nikolay B. Pestov, Ekaterina A. Orlova, Nickolai A. Barlev, Nadezhda M. Kolyasnikova, Aydar A. Ishmukhametov

**Affiliations:** 1Laboratory of Tick-Borne Encephalitis and Other Viral Encephalitides, Chumakov Federal Scientific Center for Research and Development of Immune-and-Biological Products of Russian Academy of Sciences, Moscow 108819, Russia; 2Institute of Translational Medicine and Biotechnology, Sechenov First Moscow State Medical University, Moscow 119991, Russia

**Keywords:** oncolytic viruses, flavivirus, immunotherapy, viral vector, recombinant strain, cancer immunotherapy, antitumor therapy

## Abstract

Oncolytic viruses offer a promising approach to tumor treatment. These viruses not only have a direct lytic effect on tumor cells but can also modify the tumor microenvironment and activate antitumor immunity. Due to their high pathogenicity, flaviviruses have often been overlooked as potential antitumor agents. However, with recent advancements in genetic engineering techniques, an extensive history with vaccine strains, and the development of new attenuated vaccine strains, there has been a renewed interest in the Flavivirus genus. Flaviviruses can be genetically modified to express transgenes at acceptable levels, and the stability of such constructs has been greatly improving over the years. The key advantages of flaviviruses include their reproduction cycle occurring entirely within the cytoplasm (avoiding genome integration) and their ability to cross the blood–brain barrier, facilitating the systemic delivery of oncolytics against brain tumors. So far, the direct lytic effects and immunomodulatory activities of many flaviviruses have been widely studied in experimental animal models across various types of tumors. In this review, we delve into the findings of these studies and contemplate the promising potential of flaviviruses in oncolytic therapies.

## 1. Introduction

Given the expanding array of immunotherapeutic agents targeting various tumor types, recent research efforts have been channeled towards discovering novel treatments that can further amplify the efficacy of antitumor therapy using oncolytic viruses (OV) [1]. One strategy involves leveraging existing infectious disease vaccines as a supplement to immunotherapy. This cost-effective and potentially swifter method could mitigate the high costs of current cancer drugs [2]. Another strategy employs genetic engineering techniques to augment the oncolytic properties of viruses, yielding improved strains [3], especially using synthetic biology approaches [4]. These advancements have considerably accelerated the progress of virotherapy.

Several phylogenetically diverse viruses from different families have shown promising results in numerous preclinical studies, and the most popular OV platforms in clinical trials for various cancer types belong to the families *Herpesviridae*, *Adenoviridae*, *Poxviridae*, and *Parvoviridae* among DNA viruses and *Paramyxoviridae*, *Picornaviridae*, *Reoviridae*, *Retroviridae*, and *Rhabdoviridae* among RNA viruses [1,5].

As a form of immunotherapy, virotherapy has already demonstrated promising outcomes in patients with a poor prognosis of tumor progression, for example, in glioblastoma [6,7]. A number of recent reviews on OVs can be recommended with the focus on diversity of OVs [1,5,8,9,10], country-specific experience (for example, in Russia and USSR [11], China [12], Germany [13], and Japan [14]), challenges of clinical trials [15], as well as imperfections of preclinical models [16]. T-VEC, an anti-melanoma HSV-1 based OV with granulocyte colony-stimulating factor (G-CSF) transgene, holds the distinction of the most extensive clinical experience, hence, a review on this particular OV is especially interesting [17].

Despite the breakthrough results of new therapies using immune checkpoint inhibitors (ICI), lack of response, resistance, and toxicity in some patients remain key obstacles to their widespread use [18]. Tumors sensitive to ICI are considered immunologically “hot”, with high levels of TILs (tumor infiltrating lymphocytes), increased expression of PD-L1 (programmed cell death-ligand 1), and a high mutation load [18,19]. ICI-insensitive tumors are considered immunologically “cold” due to lack of expression or presentation of tumor-associated antigens (TAA), low levels of TILs, infiltration by immunosuppressive cells (Tregs, M2 macrophages, myeloid suppressor cells, and neutrophils), expression of inhibitory cytokines (e.g., TGF-β and IL-10), or overexpression of alternative immune checkpoints (such as TIM-3, LAG-3, BTLA, and VISTA), respectively, and the use of ICI in the therapy of such tumors is not effective [1,19,20].

OVs, when used to eradicate tumor cells, concurrently stimulate the antitumor immune response through so-called immunogenic cell death (ICD). This promotes the transition of “cold” tumors to “hot” tumors, thereby potentially increasing the count of ICI-sensitive cancers and enhancing the effectiveness of cancer immunotherapy [21].

Advances in genetic methodologies and the emergence of new attenuated strains have opened the possibility of considering even highly pathogenic viruses, such as those from the Flavivirus genus, as donors of useful oncolysis-promoting elements. Recent studies have demonstrated that well-characterized vaccine candidates based on Zika virus strains (ZIKV), which have previously shown genetic stability and safety in animal models, exhibit oncolytic activity against glioblastoma stem cells (GSCs) [22,23]. One in vitro study showed that a CpG-encoded variant of Zika virus displayed slower infection kinetics in nonmalignant brain cells, but high infectivity and oncolytic activity in GSCs, and the virus also replicated effectively in tumors originating from GSC, with a significant reduction in tumor growth [24]. Antitumor efficacy has also been described for other flaviviruses, which will be discussed below. In this review we analyze the oncolytic potential of viruses from the Flavivirus genus and successful genetic modifications thereof to improve their safety and efficacy as OVs.

## 2. Description and Life Cycle of Flaviviruses

The Flavivirus genus includes more than 70 arboviruses that are transmissible and are carried predominantly by mosquitoes and ticks. Some members of the Flavivirus genus are highly pathogenic to humans and cause such diseases as yellow fever, dengue fever, West Nile fever, Zika fever, Japanese encephalitis, tick-borne encephalitis, etc. [25]. The pathogenesis of many flaviviruses is linked to their replication in cells of the nervous system. Some flaviviruses are particularly intriguing due to their neurotropism and penetration through the blood–brain barrier (BBB). However, both natural and genetically modified flaviviruses can be viewed as potential OVs.

The flavivirus genome is represented by capped positive strand ssRNA of about eleven thousand nucleotides with a single open reading frame. Virions consist of an icosahedral nucleocapsid formed by genomic RNA and capsid protein C. The nucleocapsid is surrounded by a lipid envelope calls the supercapsid, which contains glycosylated protein E and membrane protein M [26].

Flaviviruses enter cells via receptor-mediated endocytosis, which involves protein E interacting with receptors in the target cells [27], followed by nucleocapsid separation and release of viral RNA into the cytoplasm. The subsequent ribosomal translation of the viral RNA gives the polyprotein, which is co-translationally and post-translationally cleaved by cellular and viral proteases into three structural (C, prM, and E) and seven nonstructural proteins (NS1, NS2a, NS2b, NS3, NS4a, NS4b, and NS5). The NS2a, NS4a, and NS4b proteins are involved in the formation of the replicative complex. The NS1 protein is believed to stabilize the replication complex on the membrane by linking the NS4a and NS4b proteins together, thus providing a scaffold for replication [28]. The NS3 protein plays a role in the processing of viral polyproteins. Composed of two domains, the NS5 protein includes a domain functioning as a viral RNA-dependent RNA polymerase and a methyltransferase domain involved in modifying viral cap structures. NS5 facilitates the replication of genomic RNA that initiates with the synthesis of a negative RNA strand subsequently used as a template for “genomic” RNA synthesis. The assembly of virions takes place in the endoplasmic reticulum (ER), where viral proteins and genomic RNA amalgamate to form immature viral particles. In the process of budding from the ER, the immature viral particle receives a lipid envelope and undergoes protease- and pH-dependent maturation during its movement from the ER through the trans-Golgi network to the cell surface. During maturation, protein E undergoes significant reorganization, while protein prM is transformed into M protein, thereby leading to complete maturation of virions that leave the cell during exocytosis [29] (Figure 1).

While there is a wealth of knowledge on flavivirus reproduction, many molecular interactions between flaviviruses and their target cell receptors, as well as the nature of these receptors, are not fully understood. Initially, flaviviruses seem to adhere to the target cell surface using glycosaminoglycans (GAG) of heparan-sulfate proteoglycans or syndecans. Glycosaminoglycans, present on the surface of all tissue cells, are frequently exploited by various viruses for attachment [30].

Once the flavivirus virions bind to the cell surface, the virus’s entry is facilitated by cellular receptors different from glycosaminoglycans. While there is considerable conjecture about the receptors flaviviruses use for entry, the precise role of these receptors in endocytosis is ambiguous. Research indicates that Dengue, West Nile, and Japanese encephalitis viruses may use C-type lectin receptors. These receptors are abundantly expressed on monocytes, macrophages, and dendritic cells (DCs), playing an important role in immune response activation. Other studies have pointed to the participation of receptors such as TAM tyrosine kinases, phosphatidylserine receptors, high-affinity laminin receptor, Prohibitin, Integrin αvβ3, SRB1, Claudin-1, and NKp44 in flavivirus reproduction [31]. Given the diverse array of receptors that flaviviruses can use, the specificity of their reproduction is also dependent on intracellular factors, which remain insufficiently studied.

**Figure 1 viruses-15-01973-f001:**
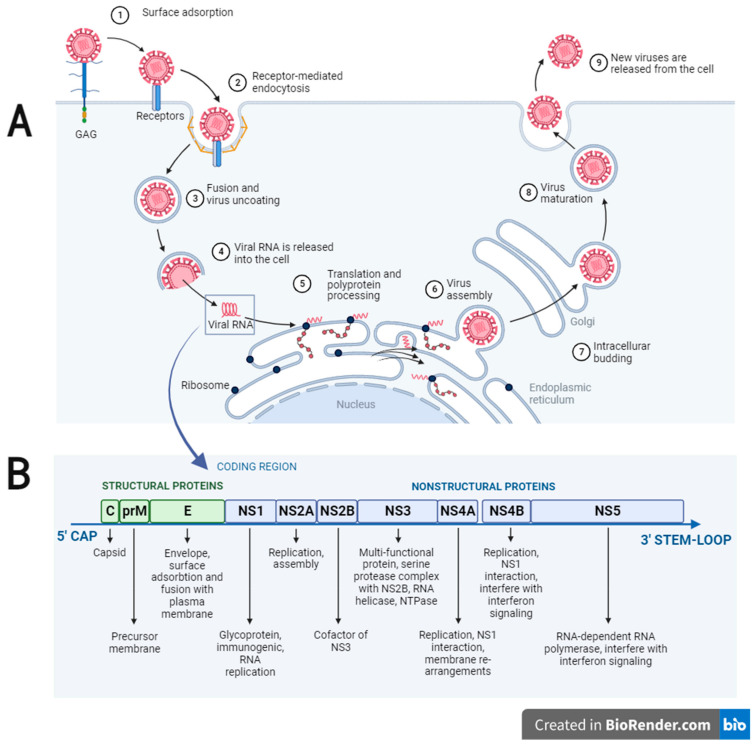
**Panel** (**A**)—**Life cycle of flaviviruses.** (**1**)—Attachment to cell surface using glycosaminoglycans (GAG), (**2**)—virus entry into the host cell via receptor-mediated endocytosis, (**3**,**4**)—release of the genomic RNA into the cytoplasm through membrane fusion mediated by viral glycoproteins, (**5**)—translation and polyprotein processing, (**6**)—immature particles pass through the Golgi, (**7**)—intracellular budding, (**8**)—virus maturation, and (**9**)—virus release. **Panel** (**B**)**—The flavivirus genome** is represented by capped ∼11 kb positive strand ssRNA with a single open reading frame. The genomic RNA is translated as a single polyprotein subsequently cleaved into three structural proteins (C—capsid, prM/M—precursor membrane, and E—envelope) and seven nonstructural (NS) proteins responsible for coordinating viral replication, assembly, and modulation of host defense mechanisms [28,32].

## 3. Flaviviruses as Oncolytic Viruses

The first works on the use of flaviviruses for virotherapy of tumors date back to the early 1950s of the 20th century. Tick-borne encephalitis virus (TBEV) was shown to inhibit the growth of various tumor types in mice, such as MCI fibrosarcoma, C1300 neuroblastoma, EO 771 mammary adenocarcinoma, Ridgeway osteogenic sarcoma, T241 sarcoma, and 180 [33,34]. Flaviviruses, including tick-borne encephalitis (TBEV), Scottish sheep encephalomyelitis (Louping ill) virus, Langat virus, Kyasanur Forest disease virus, and Omsk hemorrhagic fever virus were subsequently demonstrated to exert cytopathogenic effects in HeLa cells [35], inhibiting their growth both in vitro and in vivo [36,37,38]. St. Louis encephalitis virus (SLE) increased mean survival time of infected mice and caused regression of Ehrlich tumor in 20% of mice [39,40]. Also, in the 1960s, the Langat and Kyasanur viruses were studied for their oncolytic potential. There were no instances of a complete cure in patients following a single injection of the virus. However, in 2 out of 10 cases, there was a noticeable improvement in the overall condition of patients with leukemia, and in 4 out of 18 cases, transient benefits were recorded in patients with solid tumors of various origins. The most common adverse effect was fever. Unfortunately, a patient with bronchial cancer exhibited signs of encephalitis in response to the therapy [41]. The high pathogenicity of flaviviruses in animals and humans, the suboptimal results of their oncolytic activity when used as monotherapy, and the emergence of other more convenient and safer OVs have long hindered research into the use of flaviviruses as oncotherapeutic agents.

Renewed interest in flaviviruses for oncoviral therapy is related to research on the use of attenuated flaviviruses and the possibility of producing genetically modified variants including those carrying transgenes. Thus, attenuated Zika, yellow fever, West Nile, and Japanese encephalitis viruses proved effective for direct oncolytic action on a number of animal and human tumors and for the induction of antitumor immunity [42,43,44,45] (Table 1).

The ChimeriVax vaccine technology exemplifies an innovative approach in creating vaccines. This technology produces chimeric (recombinant attenuated) yellow fever virus (YFV), strain 17D, by substituting its prM-E genes with analogous genes from other flaviviruses. So far, this method has been harnessed to develop several chimeric flaviviruses. These have demonstrated safety in non-human primates, with some advancing through clinical trials to become registered vaccines:

➢The Dengvaxia^®^ vaccine, built on this technology, has been licensed in 20 countries following successful preclinical trials in non-human primates and human clinical trials. Its safety is comparable to that of YFV-17D. However, DENV seronegative individuals who get vaccinated and later contract DENV might be at an increased risk of severe Dengue course due to antibody-dependent enhancement of infection [46,47,48]. Similarly, chimeric DENVax viruses containing the prM and E genes of DEN-1, DEN-3, and DEN-4 were successfully created [49]. This vaccine has completed preclinical testing on Macaca fascicularis, demonstrating promising safety, and is now undergoing phase 3 clinical trials [47]. Another technique to produce attenuated flavivirus strains involves deletions in the 3′-UTR [50];➢Remarkably, this vaccine shows cased the most impressive safety record among all vaccines developed using this technology, even surpassing YFV-17D itself. Even after intracerebral administration in non-human primates, there were no indications of encephalitis. This vaccine has since been registered and is now in use [51,52];➢This stands as the sole candidate vaccine against the West Nile virus to clear Phase II clinical trials. On Macaca fascicularis, its safety profile mirrors that of YFV-17D [53,54];➢ChimeriVax-Zika, in comparison to the above vaccines, has only recently begun development and its positive safety profile can currently only be inferred from studies in mouse models [55]. However, a ChinZIKV vaccine was developed based on JEV LAV using a similar technique of replacing prM-E JEV genes with similar ZIKV genes. Tests in mice and rhesus macaque models demonstrated its safety, with no placental or fetal damage observed in pregnant mice infected with ZIKV after immunization with ChinZIKV [56].

Thus, a sufficient number of human nonpathogenic flavivirus strains currently exist or can be obtained, and their oncolytic efficacy should be evaluated in the future. It should also be noted that flavivirus vaccines are usually highly immunogenic [57], which may contribute to the formation of antitumor immunity when these viruses are used as OVs.

### 3.1. Zika Virus

The most studied of the oncolytic flaviviruses is the Zika virus (ZIKV). Zika virus is capable of crossing the placental barrier, infecting the fetus, and causing neurodevelopmental disorders, including microcephaly in newborns. However, in adult patients, Zika virus infection is usually asymptomatic, with less than 20% of patients reporting mild fever, rash, and joint pain for about 7 days [58]. A number of studies have confirmed that Zika virus selectively infects fetal neural stem cells (NSCs) [59], which prompted scientists to investigate Zika virus as an OV against brain tumors. It has been suggested that AXL serves as a receptor for Zika virus entry. AXL is a cell surface receptor tyrosine kinase, a member of the TAM family of kinases whose expression is significantly increased in various types of malignancies [60]. This assumption explained the specificity of Zika virus towards neurons; however, other studies have shown that the presence of AXL is not necessary for Zika virus entry into the cell [61]. As a result, the role of AXL in the life cycle of Zika virus remains controversial. In studies [62,63] it was shown that the expression of the cell surface receptor CD24 is required for the oncolytic action of Zika virus. Restoration of CD24 expression in the Zika virus–immune neuroblastoma cell line SK-N-AS increased the oncolytic potential of Zika virus compared to its effect on the CD24-defective cell line. Also, it was noted that the Zika virus NS5 protein, unlike other nonstructural proteins, reduces glioma cell proliferation, migration and invasion in vitro, and promotes survival of C57/BL6 mice with intracranial glioma [64].

Several factors render the Zika virus appealing as an OV. Firstly, selectivity for malignant cells may be linked to the virus’s sensitivity to type 1 interferon [22], thus Zika virus targets predominantly cells with this signaling pathway disrupted during tumor development. Secondly, the Zika virus is able to suppress apoptosis [65,66,67], and this feature may be beneficial for induction of ICD. Lastly, the potential for a delayed cytolytic effect of the Zika virus on tumor cells could be crucial in counteracting the immunosuppression induced by the tumor environment [68,69,70].

Recently, a number of remarkable achievements in the oncolytic activity of Zika virus in vivo have been reported. One study [22] demonstrated that a mouse-adapted strain of ZIKV-Dakar can increase the survival of C57BL/6 mice with GL261 and CT-2A glioblastomas with a decrease in tumor volume and the absence of neurological symptoms from the virus. In a study of the Brazilian Zika virus, strain ZIKV^BR^ for medulloblastoma and rhabdoid tumor therapy, tumor regression, increased survival, and complete tumor regression was observed in 7 of 29 mice [71]. A study was also conducted on immunocompetent dogs with brain tumors, where Zika virus therapy showed an improvement in neurological symptoms and a decrease in tumor volume. In addition, two dogs showed an increase in the number of immune cells in the tumor microenvironment [72]. The use of the Zika virus vaccine strain ZIKV-LAV for treating glioblastomas in immunodeficient mice led to increased survival rates and slowed tumor growth. Moreover, this strain exhibited a commendable safety profile; even when administered intracerebrally to mice, no neurological symptoms were reported. It was determined that Zika virus infection triggers the tumor necrosis factor (TNF) signaling pathway, which suggests that ZIKV-LAV treatment may not only cause direct cell death but also activate antitumor immunity.

It was observed that the Zika virus’s oncolytic activity against glioblastomas is linked to the expression of SOX2 in these tumors [23]. This leads to the suppression of the interferon signaling pathway, thereby facilitating Zika virus replication [73]. SOX2 is a major regulator of antiviral response and apoptosis and is highly expressed in glioblastoma cells, which explains the predominant lytic action of Zika virus. In addition, SOX2 plays an important role in the regulation of αV integrin expression. Integrin αV forms heterodimers, predominantly with integrin β5, whose blockade using specific antibodies leads to a decrease in the Zika-virus-induced oncolytic action on glioblastoma cells. Thus, the SOX2-integrin αVβ5 appears to be crucial for the oncolytic activity of Zika virus against glioblastoma.

In C57BL6/J mouse models with CT-2A glioblastoma, the Zika virus has been shown to promote intratumor infiltration of CD4+ and CD8+, thereby altering the tumor microenvironment and promoting antitumor action [69]. Zika virus infection intensifies the infiltration of cytotoxic lymphocytes into the tumor, with the activation of CD8+ T sustained over an extended period. These lymphocytes offer protection to mice, when reintroduced into syngeneic tumors. Furthermore, Zika virus therapy encourages the activation of the type I interferon signaling pathway in glioblastoma cells, heightens the sensitivity of glioblastoma to PD-L1 immune checkpoint blockade, and also prolongs the survival of mice while inhibiting tumor growth [69,74].

It should be emphasized that the Zika unstructured protein NS2A was shown to inhibit the tumor suppressive p53 functions in the cells, arguing that the effect of Zika virus-mediated oncolysis may be cell context-dependent. P53 is considered a guardian of the human genome and hence senses viral infections as potential threat to integrity of the genome [75]. p53 is a potent activator of apoptosis, and therefore, its activity and the protein level are kept in check with a number of ubiquitin ligases, including Mdm2, Pirh2, etc [76,77]. These E3 ligases mediate post-translational ubiquitination of p53 targeting it for proteasomal degradation, being the transcriptional targets of p53 themselves. Both E3 ligases and the proteasome are subjected to post-transcriptional modifications themselves [78,79]. Not surprisingly, the NS2A protein can attenuate ATM-dependent phosphorylation of p53 and its nuclear import upon transfection into p53-positive osteosarcoma or glioma cells [75].

### 3.2. Yellow Fever Virus

The attenuated 17D strain of the yellow fever virus may also be a promising candidate for oncoviral therapy. This strain has shown a reliable safety profile in immunocompetent children and adults, and is routinely used for prophylactic purposes to prevent disease in endemic regions [80]. Given its active use in vaccine prophylaxis, it is plausible that its intratumor administration could manifest immune-mediated effects against the tumors. 17D yellow fever vaccine (Stamaril, Sanofi Pasteur) exhibited oncolytic properties in B16-OVA melanoma and MC38 colon carcinoma models [42]. Even though administration of the virus has not led to a complete cure in either case, a noticeable delay in tumor progression was observed. In addition to the direct impact of the virus itself, specific CD8+ cells and the induction of IFNα/β synthesis have been shown to contribute to the antitumor action of 17D. Notably, NK cells do not participate in this process. Prior immunization of mice by the 17D increased both antitumor activity and animal survival. Similar results were seen with the combined therapy using immune checkpoint blockers, anti-PD-1 and anti-CD137 [42]. McAllister A. et al. have shown that recombinant live attenuated (strain 17D) YFV constructed to express a cytotoxic T-lymphocyte epitope derived from chicken ovalbumin (OVA) works as an effective therapeutic vaccine for the treatment of murine experimental solid tumors and pulmonary metastases in B16-OVA mouse melanoma [81].

### 3.3. West Nile Virus

West Nile virus (WNV) was first used as an OV in the 1950s for patients with advanced, inoperable tumors of various types. WNV, pre-passaged in mice, did not induce clinical symptoms in recipients, and unfortunately, no therapeutic antitumor effect was observed [82]. The isolated WNV, strain Egypt 101 triggered noticeable tumor regression in 4 out of 34 tested patients and a less prominent regression in 5 patients. Most patients also exhibited virus replication in their tumor tissues. Unfortunately, encephalitis was diagnosed in some patients [83]. Following the initial trials, due to a lack of pronounced antitumor effects and high pathogenicity, WNV has not received attention as an OV for a long time. However, the emergence of attenuated WNV strains renewed interest in the West Nile virus. For instance, a live attenuated WNV-poly(A) vaccine [84] was tested on the B16F1 murine melanoma model. The WNV-poly(A) therapy resulted in the inhibition of growth and reduction in tumor volume, without causing pathological changes in healthy tissues [45]. Moreover, the activation of CD8+ cells and their contribution to the overall antitumor effect were observed. In models of human melanoma A375 xenografts, ovarian carcinoma SKOV3, and hepatoma Huh7 in BALB/c-nu/nu mice, it was shown that WNV-poly(A) therapy resulted in growth inhibition and tumor volume reduction without side effects. It is likely that the selective effect of WNV-poly(A) on the tumor is due to its sensitivity to type 1 interferon [45].

Based on Kunjin virus (KUN), which is a subtype of WNV [85], an interesting OV candidate has been constructed with transgenes inserted between protein C (AK20) and envelope protein E (AK22). The disadvantage of such a system was the low yield of infectious particles, so a specific cell line was subsequently developed to facilitate the production of the modified KUN virus [86,87]. The developed vector systems were applied for oncoviral therapy in murine models. Non-cytopathogenic virus-like KUN virus particles encoding G-CSF were locally injected into tumors of mice with implanted subcutaneous CT26 colon carcinoma and with B16-OVA melanoma [88]. A complete cure was demonstrated in more than 50% of the animals. CT26 tumor regression correlated with antitumor CD8^+^ induction, regression of CT26 carcinoma lung metastases was demonstrated, as well as protection against fresh injections of the cancerous cells. In the case of B16-OVA melanoma, four of six mice were cured completely and protected from reinfection with B16-OVA tumor. Injection of a KUN-based replicon carrying the cytotoxic T lymphocytes (CTLs) epitope protein E7 of herpes simplex virus 16 protected 80% of mice from tumor growth [89]. In these experiments, protected mice showed no tumor growth for 70 days after the initial injection of tumor cells, and mice also held antitumor immunity when the tumor cells were reintroduced. The immune response and antitumor defense were more pronounced in mice immunized with virus-like KUN particles. Using in vitro stimulation of splenocytes, it was shown that long-term immunological memory in mice immunized once with KUN replicons can lead to induction of an immune response for at least 6 months [90]. The data obtained specify the effectiveness of antitumor protection induced via KUN virus-based replicons carrying tumor antigens, indicating the formation of long-term immunological memory for tumor antigens.

In contrast to the negative effect of Zika virus on p53, West Nile virus was shown to activate p53 [91]. The WNV capsid was capable of binding to and sequestering the E3 ligase MDM2 into the nucleolus. Similar to the effect of small molecule inhibitors of Mdm2 [92,93], the WNV capsid increased the protein level of p53, thereby affecting the transcription of its target genes.

### 3.4. Japanese Encephalitis Virus

An attenuated strain of Japanese encephalitis virus (JEV), JEV-LAV has been used in China and other countries since 1989 for vaccine prophylaxis, and no cases of vaccine-associated encephalitis have been reported to date [94]. In study [43], glioblastoma cell line GL261-luc was injected into mouse brain. After tumor maturation, intratumor injection of JEV-LAV was performed. The therapy was shown to inhibit tumor growth, decrease tumor volume, increase mice survival, CD8+ tumor infiltration, and increase PDL-1 expression. Combination therapy with an anti-PD-L1 immune checkpoint blocker also showed inhibition of tumor growth and increased survival of mice. No pathologies were observed in mice after intracerebral injection of JEV-LAV [43].

### 3.5. Dengue Virus

The initial mentions of Dengue virus (DENV) as an OV can be traced back to the early 1980s: in Albino mice with Ehrlich tumors, tumor regression was observed in 20% of cases when treated with DENV, similar to the outcomes observed with St. Louis encephalitis virus (SLE) [40]. With the development of genetic engineering methods and the appearance of new attenuated strains, the Dengue virus has again attracted the attention of researchers. For example, attenuated Dengue virus PV001-DV has been studied as an oncolytic in vitro and ex vivo. In human melanoma cell lines A2058, SK-MEL-2, SK-MEL-5, and SK-MEL-28, PV001-DV was shown to induce cell death via apoptosis and increase PD-L1 expression, which may contribute to an enhanced oncolytic effect when combined with anti-PD-L1 therapy. The oncolytic effect has also been demonstrated in an ex vivo melanoma sample [95]. However, the antitumor effects of PV001-DV have not yet been confirmed in animal models.

Importantly, p53 was shown to significantly interfere with replication of Dengue virus in liver cancer HEpG2 cells [96]. On the molecular level, recently published data [97] indicate that Dengue virus may trigger ferroptosis in the target cells, which is amplified in the presence of active p53.

**Table 1 viruses-15-01973-t001:** Antitumor efficacy of flaviviruses.

Virus *	Strain	Combination Therapy	Tumor Type (Model)	Animal Model	Description	Reference
ZIKV	ZIKV-LAV	-	Glioblastoma xenografts (Orthotopic)	BALB/c mice	Safe for intracerebral administration; delayed tumor growth and increased survival rate; development of antitumor immunity	[23]
	ZIKV-Dakar	-	Mouse glioma GL261 and CT-2A (Orthotopic)	C57BL/6, C57BL6/J mice	Increase in survival; tumor regression; activation of CD8+, myeloid cells; development of antitumor immunity	[22,69]
	ZIKV-Dakar	Anti-PD-1	Mouse glioma CT-2A (Orthotopic)	C57BL6/J mice	Increased survival; tumor regression	[69]
	ZIKV-GZ01 ZIKV-FSS	-	Mouse glioma GL261 and CT-2A (orthotopic)	C57BL/6N mice	Suppression of tumor growth; increased survival; activation of CD4+ and CD8+, type 1 interferon signaling pathways; increased efficacy of immune checkpoint blockers	[74]
	ZIKV^BR^	-	Human medulloblastoma DAOY, USP13-MED, human rhabdoid tumor USP7-ATRT (orthotopic)	BALB/c mice	Increased survival rate; tumor regression	[71]
		-	CNS tumor (spontaneous)	Pit Bull, Boxer, Dachshund Dogs	Improvement of neurological symptoms; reduction in tumor volume	[72]
YFV	17D-pOva	-	B16-Ova, B16F0 mouse melanoma, pulmonary metastasis	C57BL/6 mice	Delayed tumor growth; increased survival, reduced both the size and number of lung metastases; induction of Ova-specific CD8+ T cells	[81]
	17D	Anti-PD-1 Anti-CD137	B16-OVA mouse melanoma, MC38 mouse colon carcinoma (subcutaneous)	C57BL/6 mice	Delayed tumor growth; increased survival; activation of IFN I, CD8+	[42]
WNV	WNV-poly(A)	-	Mouse melanoma B16F1 (subcutaneous)	C57BL/6 mice	Delayed tumor growth; decrease in tumor volume; CD8+ activation	[45]
		-	Human melanoma A375, human ovarian carcinoma SKOV3, human hepatoma Huh7 (subcutaneous)	BALB/c-nu/nu mice		
	KUN	-	CT26 mouse colon carcinoma, B16-OVA mouse melanoma (subcutaneous)	BALB/c, C57BL/6 mice	Regression of tumors and metastases; CD8+ induction; development of antitumor immunity	[88]
	KUN with E7 epitope	-	Tumor cell lines EL4.A2 and TC-1 (subcutaneous)	BALB/c mice	Development of antitumor immunity to herpes simplex virus 16 protein E7	[89]
JEV	JEV-LAV JEV-LAV	Anti-PD-1	Mouse glioma GL261-luc (orthotopic)	C57BL/6J mice	Inhibition of tumor growth; reduction in tumor volume; increase in lifetime, CD8+ infiltration; increase in PDL-1 expression	[43]
TBEV	Far East	-	MCI mouse fibrosarcoma, C1300 mouse neuroblastoma, EO 771 mouse mammary adenocarcinoma, Ridgeway mouse osteogenic sarcoma, T241, and 180 mouse sarcomas	C57BL mice, outbred mice	Delayed tumor growth	[33,34]
SLE	-	-	Mouse Sarcoma 180(subcutaneous)	White swiss mice	Increased survival	[39]
	Parton	-	Ehrlich’s tumor (intraperitoneal)	Albino mice	Tumor regression in 20% of cases	[40]
DENV	TR1751	-				

* ZIKV—Zika virus, YFV—Yellow Fever Virus, WNV—West Nile virus, KUN—Kunjin virus, JEV—Japanese encephalitis virus, TBEV—Tick-borne encephalitis virus, SLE—St. Louis encephalitis virus, and DENV—Dengue virus.

## 4. Discussion

Oncotherapeutic means based on flaviviruses hold promise, since such OVs manifest their antitumor action both through fast and direct lysis of tumor cells and induction of immune-mediated reactions to the tumor cells. These actions may be improved further with the construction of recombinant OVs capable of introducing sequences that encode tumor suppressors, ICD-promoting proteins, and cytokines, which enhance the immune response against tumor antigens, into the tumor cells.

The ability of flaviviruses to cross the BBB can be employed to treat brain tumors such as glioblastomas [23]. In this case, the virus can be administered systemically into the bloodstream; however, high immunogenicity of flaviviruses [98,99,100,101] may hinder effective delivery Firstly, if multiple doses of the virus need to be administered, there is a high likelihood of the immune system inactivating the virus upon re-administration. Secondly, inactivation of the virus can occur in individuals who have previously received a vaccine against a similar flavivirus or have been affected by a specific flavivirus-related disease.

Additionally, adverse reactions resulting from antibody-dependent enhancement of infection may arise when a different flavivirus is encountered compared to the one that prompted immunization (a well-known phenomenon in DNV infection). To overcome this limitation, one approach is to encapsulate viral particles in biodegradable nanoparticles like liposomes or employ cell carriers such as monocytes or dendritic cells. Another strategy involves coating the viral particle with various polymers such as polyethylene glycol [102,103].

On the other hand, the immunogenicity of flaviviruses can augment the antitumor effect when administered intratumorally. In such cases, flaviviruses have the potential to remodel the immunosuppressive tumor microenvironment and foster a pro-inflammatory milieu. Moreover, flaviviral OVs may be even more effective when the organism is pre-immunized against these flaviviruses by corresponding vaccines.

Both live and inactivated flavivirus vaccines can activate T-cell immunity: infection and subsequent induction of humoral and cellular immune responses (through cytokine-mediated recruitment of neutrophils, natural killer cells, macrophages, and T-lymphocytes) the immunogenicity of tumor cells can be heightened to overcome the local immunosuppression established by tumors [68,70].

Unlike certain other promising OVs, one notable characteristic of flaviviruses is the availability and extensive historical use of commercially available and effective flavivirus vaccines. These vaccines are derived from both attenuated viral strains and strains with high pathogenicity for humans [57,104]. These vaccines have proven their effectiveness and safety by immunizing hundreds of millions of individuals [105,106].

Compared to the production of recombinant adenoviruses, flavivirus vaccines production has a longer history, is better studied, and the vaccines themselves are more diverse, so their production is easier and more cost-effective. The technology of flavivirus vaccines production is well developed, and the cell cultures used are thoroughly studied [107].

In conclusion, while no flaviviral OVs are currently in registered clinical trials, their potential remains enormous. Attenuated chimeric flaviviruses, which were originally developed as flavivirus vaccines and have already demonstrated their safety in vivo, might find renewed interest in oncovirotherapy. Their success hinges on addressing challenges related to side effects, transgene stability, and intricate immune system interactions.

## Data Availability

Not applicable.

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
