# Peer review of "Flaviviruses in AntiTumor Therapy"

_viruses, 2023, doi:10.3390/v15101973_

Round 1

Reviewer 1 Report

The flaviviruses are single-stranded RNA viruses that replicate only in the cytoplasm, avoid the risk of integration into the cell genome, and have good safety after attenuated viruses or genome modification. This manuscript has good innovation and can arouse readers' interest. There are some improvements for this manuscript.

1. There are some spelling mistakes througout the manscipt, and the language and grammar of this manuscript shoud be improved. 

2. The description in Figure 1 should be more detailed, with the genome labeled A and the life cycle labeled B.

3. What are the receptors of Flaviviruses in target cell membrane? The authors should descibe them in the manuscript and Figure 1.

4. Table 1 shoud add the abbreviation of different brief virus names.

5. The clinical trials using Flaviviruses as oncolytic virus drugs should be added in the main text and listed as a another table.

6. In the part of discussion, the authors points that the immunogenicity of flaviviruses may hinder effective treatment through three ways. But they only gave two possible solutions. The author should discuss possible approaches in more detail and describe them with specific examples.

The English language is reasonable and readable. There is only a few grammar and spelling mistakes that are needed to improve.

Author Response

Dear Reviewer,

We found all remarks and suggestions highly beneficial for our work and tried to correct the manuscript accordingly.

Point by point answers are appended below:

The flaviviruses are single-stranded RNA viruses that replicate only in the cytoplasm, avoid the risk of integration into the cell genome, and have good safety after attenuated viruses or genome modification. This manuscript has good innovation and can arouse readers' interest. There are some improvements for this manuscript.

1) There are some spelling mistakes througout the manscipt, and the language and grammar of this manuscript shoud be improved.

We have rechecked the text for typos and grammatical errors.

2) The description in Figure 1 should be more detailed, with the genome labeled A and the life cycle labeled B.

Thanks for this suggestion. We included this labeling of the panels.

3) What are the receptors of Flaviviruses in target cell membrane? The authors should descibe them in the manuscript and Figure 1.

We included two paragraphs devoted to this interesting problem

4) Table 1 shoud add the abbreviation of different brief virus names.

We added an abbreviation list in the table’s bottom.

 5) The clinical trials using Flaviviruses as oncolytic virus drugs should be added in the main text and listed as a another table.

In the revised manuscript, we have stated that apparently no registered trials of flaviviruses for oncological purposes can be found.

6) In the part of discussion, the authors points that the immunogenicity of flaviviruses may hinder effective treatment through three ways. But they only gave two possible solutions. The author should discuss possible approaches in more detail and describe them with specific examples.

 We added some refernces and discussion on this point.

Comments on the Quality of English Language. The English language is reasonable and readable. There is only a few grammar and spelling mistakes that are needed to improve.

We have rechecked the text for typos and grammatical errors.

Reviewer 2 Report

In the provided mainly well written review, which is devoted to an interesting topic of using as Flaviviruses as oncolytic agents, there are several shortcomings, the elimination of which will significantly improve the text. In the attached file, I list the shortcomings, as well as ways to eliminate them and improve the text.

In the attached file, I wrote that the discussion section language and logic needs to be improved first of all. The section has absolutely no structure and the text needs to be divided into paragraphs. In the table in the "Description" column, punctuation marks are missing and capital letters are either present or absent and do not correspond to punctuation marks. The sentence corresponding to line 138 is difficult to understand. It's worth rewriting.

Author Response

Dear Reviewer,

We found all remarks and suggestions highly beneficial for our work and tried to correct the manuscript accordingly.

Point by point answers are appended below:

In the provided mainly well written review, which is devoted to an interesting topic of using as Flaviviruses as oncolytic agents, there are several shortcomings, the elimination of which will significantly improve the text. In the attached file, I list the shortcomings, as well as ways to eliminate them and improve the text.

Comments on the Quality of English Language. In the attached file, I wrote that the discussion section language and logic needs to be improved first of all. The section has absolutely no structure and the text needs to be divided into paragraphs. In the table in the "Description" column, punctuation marks are missing and capital letters are either present or absent and do not correspond to punctuation marks. The sentence corresponding to line 138 is difficult to understand. It's worth rewriting.

                               We tried to adress these issues as follows:

  1. The review would benefit from the addition of a list of abbreviations.

We included an abbreviation list

  1. Introduction

Lines 60-63. 24 references are supporting information from one sentence. In scientific literature, the use of references is meant to support and provide evidence for the points you are making, rather than overwhelming a sentence or thought with an excessive number of references. Having 24 references for a single sentence or thought is certainly problematic. Select Key References: Choose a few high-quality references, preferably reviews that directly support your point and are highly relevant to the families of oncolytic viruses you are writing about. Integrate References Smoothly: and provide context for the reader. Use Supporting Sentences: If you have multiple references that are relevant to a particular point, consider spreading them out across supporting sentences or paragraphs rather than cramming them into a single sentence. Provide Context: Briefly explain the relevance of the references you do use to ensure that the reader understands why they've been included.

We tried to improve this section by reducing the number of refernces, and adding explanations to the references so that to avoid the impression thay these references are piled together.

  1. Section: Flaviviruses as oncolytic viruses.

Line 138-141. The clarity of the sentence could be enhanced through some revisions.

                               We have this sentence rewritten.

Line 147. Please add information that safety was tested using non-human primate animal model.

                               We have added this information wherever possible.

  1. Table 1.
  2. A) Some fragments of text in the column “Description” lacks punctuation marks. Some fragments of phrases begin with capital letters, while others do not. A consistent style and the use of punctuation marks are necessary. Without them, the information presented in the table becomes challenging for the reader to comprehend.

                               We tried to fix these typos completely.

  1. B) In the same table, there is a column called “animal models”. In this column there is a description of clinical cases. It is about the Egypt 101 virus variant and reference [80].

                               Since there is only one clinical study, we removed this case from the table.

  1. C) In the same table there is an expression: mice et al. This is in the sentence with references 33, 34. It seems that the authors of the article are mice and colleagues.

                               Thanks, it is really funny, we corrected this.

  1. D) On page 10, the table format is modified, and the column format is broken.

                               We tried to repair this.

  1. Discussion: In this section, the content of the text and the form of writing are the weakest. One gets the feeling that the authors wrote this text in a hurry. While it is worth noting that this section is very important for understanding the content of the review. To improve the text in this section, I would advise the authors to first break up the text into paragraphs, following the golden rule of one thought, one paragraph. Secondly, think about how best to structure the text. Clarity and organization can be significantly improved: The text covers a range of concepts related to flaviviruses and their potential as OVs. However, the content feels somewhat disjointed and lacks a clear flow. Line 333. The sentence at line 333 suggests that the use of flaviviruses is promising because they can have a dual role in cell lysis, either direct or immune mediated. But many other oncolytic viruses have this advantage, and flaviviruses are not exceptional by these criteria. It must be said about the dual potential role of flaviviruses in the context of other families. Later in the text, multiple ideas are presented simultaneously, lacking structure and coherence. Consequently, the text becomes challenging to comprehend. I recommend that the authors carefully consider listing all the ideas they intend to convey in this discussion. Subsequently, they should devise a strategy for presenting these ideas within distinct, yet logically interconnected paragraphs.

We modified this section by introducing paragraphs, inserting additional refernces and complete rewriting of some portions.
